



**Mapping long-term and high-resolution global gridded photosynthetically active**
**radiation using the ISCCP H-series cloud product and reanalysis data**
Wenjun Tang[1], Jun Qin[2], Kun Yang[3], Yaozhi Jiang[3]
1. National Tibetan Plateau Data Center (TPDC), State Key Laboratory of Tibetan
Plateau Earth System, Environment and Resources (TPESER), Institute of Tibetan
Plateau Research, Chinese Academy of Sciences, Beijing 100101, China.
2. State Key Laboratory of Resources and Environmental Information System, Institute
of Geographic Sciences and Natural Resources Research, Chinese Academy of
Sciences, Beijing 100101, China.
3. Department of Earth System Science, Ministry of Education Key Laboratory for
Earth System Modeling, Institute for Global Change Studies, Tsinghua University,
Beijing 100084, China.
Corresponding author and address:
Dr. Wenjun Tang
Institute of Tibetan Plateau Research, Chinese Academy of Sciences
Building 3, Courtyard 16, Lin Cui Road, Chaoyang District, Beijing 100101, China
Email: tangwj@itpcas.ac.cn
Tel: +86-10-84097046
Fax: +86-10-84097079



**Abstract:** Photosynthetically active radiation (PAR) is a fundamental physiological
variable for research in the ecological, agricultural, and global change fields. In this
study, we produced a 35-year (1984–2018) high-resolution (3 h, 10 km) global gridded
PAR dataset using an effective physical-based model. The main inputs of the model
were the latest International Satellite Cloud Climatology Project (ISCCP) H-series
cloud products, MERRA-2 aerosol data, ERA5 surface routine variables, and MODIS
and CLARRA-2 albedo products. Our gridded PAR product was evaluated against
surface observations measured at seven experimental stations of the SURFace
RADiation budget network (SURFRAD), 42 experimental stations of the National
Ecological Observatory Network (NEON), and 38 experimental stations of the Chinese
Ecosystem Research Network (CERN). Instantaneous PAR was validated against
SURFRAD and NEON data; mean bias errors (MBE) and root mean square errors
(RMSE) were, on average, 5.8 W m$^{-2}$ and 44.9 W m$^{-2}$, respectively, and correlation
coefficient ($R$) was 0.94 at the 10 km scale. When upscaled to 30 km, the errors were
markedly reduced. Daily PAR was validated against SURFRAD, NEON, and CERN
data, and the RMSEs were 13.2 W m$^{-2}$, 13.1 W m$^{-2}$, and 19.6 W m$^{-2}$, respectively at the
10 km scale. The RMSEs were slightly reduced when upscaled to 30 km. Compared
with the well-known global satellite-based PAR product of the Earth's Radiant Energy
System (CERES), our PAR product was found to be a more accurate dataset with higher
resolution.       This       new       dataset       is       now       available
at https://doi.org/10.11888/RemoteSen.tpdc.271909 (Tang, 2021).
**Keywords**: PAR; Dataset; High-resolution; Long-term



## 1. Introduction

Plants rely on chlorophyll to absorb solar radiation in the visible wavelength range

(400–700 nm) for photosynthesis (Huang et al., 2020), and sunlight in this band is
commonly referred to as photosynthetically active radiation (PAR). Thus, PAR is the
source of energy for biomass formation and may directly affect the growth,
development, yield, and product quality of vegetation (Zhang et al., 2014; Ren et al.,
2021), modulating energy exchange between Earth's surface and the atmosphere
(Zhang et al., 2021). Therefore, a high-quality PAR dataset is indispensable for studies
of ecosystems, agriculture, and global change (Frouin et al., 2018).

However, measurements of PAR are not routinely conducted at weather stations

or radiation stations. For example, PAR is not routinely observed at the Baseline
Surface Radiation Network (BSRN, Ohmura et al., 1998) or at the China
Meteorological Administration (CMA, Tang et al., 2013) weather/radiation stations.
Long-term PAR observations are only provided by a few ecological experimental
observation networks, such as the Chinese Ecosystem Research Network (CERN, Wang
et al., 2016), the AmeriFlux network (https://ameriflux.lbl.gov/), the SURFace
RADiation budget network (SURFRAD,
https://www.esrl.noaa.gov/gmd/grad/surfrad/), and the National Ecological
Observatory Network (NEON, https://www.neonscience.org/). To compensate for the
lack of PAR observations, a number of methods have been developed over recent
decades to estimate PAR. These methods can be roughly divided into two categories:
station-based methods and satellite-based methods (Tang et al., 2017).

Station-based methods mainly estimate PAR using other available variables

measured at stations using empirical or physical methods. Empirical methods usually
use the observed PAR and other variables to build an empirical relationship to conduct
PAR estimation. One such method is the well-known power law equation, which
usually uses the cosine of the solar zenith angle and the clearness index as inputs. The
clearness index, defined as the ratio of the solar radiation at the surface to that at the top
of the atmosphere (TOA), roughly reflects the solar light attenuation degree caused by
clouds, aerosols, water vapor, and other atmospheric compositions. A number of such
empirical methods based on the power law equation have been developed in the last
two decades (Alados et al., 1996; Xia et al., 2008; Hu et al. 2010; Hu and Wang 2014;
Yu et al. 2015; Wang et al., 2015, 2016). In addition, artificial neural network (ANN)
methods have also been used to estimate PAR from surface solar radiation (SSR) and
other meteorological variables (e.g., air temperature, relative humidity, dew point,
water vapor pressure, and air pressure) in a variety of ecosystems in China (Wang et al.,
2016). Generally, the aforementioned empirical methods can work well when calibrated
with local PAR observations, but the parameters in these methods are station-dependent
and their performance at locations where observations are not available will deteriorate.

Physical methods of PAR estimation generally consider various attenuations in the

atmosphere through parameterization approximation to complicated radiative transfer
processes. For example, Gueymard (1989a, 1989b, 2008) developed three physical
methods for the estimation of PAR, but these only work under clear-sky conditions. To
obtain all-sky PAR, Qin et al. (2012) further extended these methods to cloudy skies by
importing the measurements of sunshine duration that are usually conducted at most
meteorological stations. Tang et al. (2013) used the PAR method of Qin et al. (2012) to
estimate the daily PAR at more than 700 CMA routine weather stations, and found its
accuracy was comparable to those of local calibrated methods. Nevertheless, the PAR
method of Qin et al. (2012) can only be used to estimate daily PAR, and strictly can
only be applied at weather stations where the observation of sunshine duration is



available.

Alternatively, satellite-based methods can be used to map spatially continuous

PAR, but compared to SSR, little attention has been paid to PAR estimation using
remote sensing data (Van Laake and Sanchez-Azofeifa, 2004; Liang et al., 2006). There
are a few algorithms for estimating PAR using satellite data, and these algorithms may
be grouped into two categories: methods based on look-up tables (LUTs) based and
parameterization methods.

LUT-based methods can circumvent complicated radiative transfer calculations

(Huang et al., 2019) to estimate PAR directly from the satellite's signal by searching
pre-calculated LUTs. Since first proposed by Pinker and Laszlo (1992), several similar
LUT-based methods (Liang et al., 2006; Zhang, et al., 2014; Huang, et al., 2016) have
emerged to estimate PAR from regional to global scales with different satellite sources.
However, LUT-based methods are more vulnerable to various uncertainties due to their
"black-box" nature, and they are also difficult to port across different satellite platforms.

In contrast, parameterization methods do not rely on satellite platforms.

Essentially, they comprise a simplification of the radiative transfer processes, and thus
require various land and atmospheric products from satellite retrievals as inputs to
estimate PAR. To some extent, the accuracy of these methods depends on the accuracy
of the input data. On the other hand, the uncertainty of parameterization methods comes
mainly from the treatment of clouds; this is because the clear-sky part of the method is
relatively mature with uncertainty less than 10% compared with the rigorous radiative
transfer calculation (Huang et al., 2020). There has been little attention paid to specific
cloud parameterization for PAR estimation except for the work of Van-Laake and
Sanchez-Azofeifa (2004), Sun et al. (2017), and Huang et al. (2020). Sun et al. (2017)
used one (UV–visible) of their two broadbands (UV–visible and near infrared) model





(a physical-based parameterization scheme for the estimation of SSR), to estimate all-
sky PAR. By further considering the multiple scattering and reflection of clouds, Huang
et al. (2020) developed a more complicated cloud parameterization scheme and
combined this with the clear-sky PAR model of Gueymard (1989a) to estimate all-sky
PAR. Although their accuracies are both acceptable, there is no corresponding PAR
product currently being produced for relevant scientific research.

In the past, a few global PAR products have been developed, such as the global

gridded PAR products of the International Satellite Cloud Climatology Project (ISCCP-
PL, Pinker and Laszlo,1992), the Clouds and the Earth's Radiant Energy System
(CERES, Su et al., 2007), the Global LAnd Surface Satellite products (GLASS, Zhang
et al. 2014), the MODIS (MCD18A2 product, Wang et al., 2020), the Breathing Earth
System Simulator (BESS, Ryu et al., 2018), and a product from Hao et al. (2019) based
the observations from the Earth Polychromatic Imaging Camera (EPIC) onboard the
Deep Space Climate Observatory (DSCOVR, Burt and Smith, 2012). However, these
global PAR products are either too coarse in spatial resolution to meet refined analyses,
too low in temporal resolution to reflect daily variations, or too short in time series to
meet the demand of climate change studies. As a result, a high-resolution long-term
global gridded PAR product is urgently needed in the scientific community.

In this study, a high-resolution 35-year global gridded PAR product was developed

using an effective physical PAR estimation model, driven mainly by the latest high-
resolution ISCCP H-series cloud products, the aerosol product of the Modern-Era
Retrospective analysis for Research and Applications, Version 2 (MERRA-2) reanalysis
data, and water vapor, surface pressure, and ozone amount products of the ERA5
reanalysis data. We also evaluated the performance of our PAR product using in-situ
observations measured across three experimental observation networks in the United



States and China, and compared its performance with another common global satellite
product. The rest of the article is organized as follows. In Section 2, we introduce the
method used to map the global gridded PAR product. The input data for estimating the
global gridded PAR product, and the in-situ data for evaluating the performance of our
estimated global gridded PAR product are described in Section 3. Section 4 presents
the validation results of our global gridded PAR product and compares this with the
well-known satellite-based global PAR product of CERES. Section 5 describes data
availability, and our summary and conclusions are given in Section 6.

**2 Estimation of PAR**

The algorithm used to map global gridded PAR in this study was the
parameterization method developed by Tang et al. (2017), who combined the physical-
based clear-sky PAR model of Qin et al. (2012) and the parameterization scheme for
cloud transmittance of Sun et al. (2012). In calculating the surface PAR, the algorithm
takes into account various attenuation processes in the atmosphere, such as absorption
of water vapor and ozone, Rayleigh scattering, and absorption and scattering of cloud
and aerosol. In addition, the algorithm also considers the multiple reflections between
the surface and the atmosphere. The parametric expressions for the PAR algorithm are
all converted from the extensive radiative transfer calculations, and thus it is a physical
and efficient method that does not require calibration with ground-based observations.
The inputs of the PAR algorithm mainly include aerosol optical depth, cloud
optical depth, water vapor, ozone amount, surface albedo, and surface air pressure.
Tang et al. (2017) used the developed PAR algorithm to estimate instantaneous PAR
using the atmosphere and land products of the Moderate Resolution Imaging
Spectroradiometer (MODIS), and the estimated instantaneous PAR was evaluated



against in-situ observations collected by the SURFRAD network. It was found that this
algorithm performs better than previous algorithms and the estimated instantaneous
PAR can have a root mean square error (RMSE) of about 40 W m$^{-2}$. Therefore, we
expect good performance from our algorithm in mapping global gridded PAR.
Interested readers can refer to our earlier article (Tang et al., 2017) for further details.

**3 Data**
**3.1 Input data**
To produce a long-term (from 1984 to 2018) high-resolution global gridded PAR
product using the PAR algorithm presented above, we used input data from four
different sources.
The first source of input data was the latest level-2 H-series pixel-level global
(HXG) cloud products of the ISCCP, here referred to as ISCCP-HXG; these were
publicly available, spanned the period July 1983 to December 2018, had a spatial
resolution of 10 km, and a temporal resolution of 3 hours. The ISCCP-HXG cloud
products were produced by a series of cloud-related algorithms based on global gridded
two-channel radiance data (visible, 0.65 μm and infrared, 10.5 μm) merged from
different geostationary and polar orbiting meteorological satellites. We must bear in
mind that the 3-hour ISCCP-HXG cloud products denote instantaneous data at a given
moment every three hours, not a mean of 3 hours. We used four variables from the
ISCCP-HXG cloud products; these were cloud mask, cloud top temperature, and the
optical depths of water cloud or ice cloud retrieved based on the visible radiance. The
sky condition (clear or cloudy) of a pixel was distinguished by the cloud mask data, and
the cloud phase (liquid or ice) of a cloudy pixel was roughly determined by the cloud
top temperature. If the cloud top temperature (TC) of a cloudy pixel was greater than
or equal to 253.1 K, it was regarded as water cloud; otherwise, it was classed as ice
cloud. For more detailed information on the ISCCP-HXG cloud products, the reader
may refer to the cloud products article of Young et al. (2018).

The second source of input data was the aerosol product of the MEERA-2

reanalysis data, which can be downloaded from the Goddard Earth Sciences Data and
Information Services Center of the National Aeronautics and Space Administration
(NASA). MERRA-2 assimilates ground-observed aerosol optical depth (AOD)
measured at the AERONET (Holben et al., 1998), and satellite-retrieved AOD from the
MODIS Aqua and Terra sensors, MISR sensor, and AVHRR sensor (Randles et al.
2017). The MERRA-2 hourly aerosol product used in this study was called
"tavg1_2d_aer_Nx", having a spatial resolution of $0.5° \times 0.625°$, a temporal resolution
of 1 hour, and a time period of 1980 to present. Two variables of the MERRA-2 aerosol
product were used in this study; these were the total AOD at 550 nm and the total
aerosol Ångström parameter (470–870 nm). To map the global gridded PAR product
with a spatial resolution of 10 km, we re-sampled the MERRA-2 aerosol product to a
spatial resolution of 10 km.

The third source of input data was the routine weather variables of the ERA5

reanalysis data, which mainly included total column ozone, total column water vapor,
and surface pressure, with a spatial resolution of 25 km and a temporal resolution of 1
hour. Total column ozone and total column water vapor were used to calculate the
transmittance due to ozone absorption and water vapor absorption, respectively.
Surface pressure was used to calculated the Rayleigh scattering in the atmosphere. To
maintain consistency with the spatial resolution of the ISCCP-HXG cloud product,
these three routine weather variables of the ERA5 reanalysis data were re-sampled to
10 km.



The fourth source of input data was albedo data from the MODIS MCD43A3
product (Schaaf et al., 2002) and from the Satellite Application Facility on Climate
Monitoring (CM–SAF) (CLARA-A2-SAL, Karlsson et al., 2017), to take into account
the multiple scattering effect between the land surface and atmosphere on the
calculation of PAR. The spatial resolutions of MODIS and CM-SAF were both 5 km,
and thus we downscaled them to 10 km. The MODIS albedo product was used after
2000, the date when it first became available, and the CM-SAF albedo product was
used before 2000 (when MODIS was unavailable). The use of different albedo products
will lead to inconsistent accuracy for the final global gridded PAR product, and thus
thus caution should be exercised when performing trend analyses.

**3.2 In-situ measurements**
In-situ PAR measurements collected across three networks from the United States
and China were used to validate our global gridded PAR product. PAR measurements
at those networks are all quantified as photosynthetic photo flux density ($\mu$ mol m$^{-2}$ s$^{-1}$
$^{1}$), and McCree's conversion factor with a value of approximately 4.6 (McCree, 1972)
was used to convert the quantum units of PAR into energy units (W m$^{-2}$) of PAR. The
first network used was SURFRAD (Augustine et al., 2000) of the National Oceanic and
Atmospheric Administration (NOAA), which contains seven experimental stations
(Goodwin Greek, Fort Peek, Bondville, Desert Rock, Sioux Falls, Table Mountain, and
Penn State) in different climatic regions (red pentagrams in Fig. 1). LI-COR Quantum
sensors were used to measure PAR at the SURFRAD network. The standards of
instrument calibration for the Baseline Surface Radiation Network (BSRN) were
adopted and the quality of radiation data at SURFRAD were considered to be
comparable to those of the BSRN. Many previous studies have used SURFRAD



radiation data to evaluate their algorithms for estimation of different radiation
components. The PAR observations at 1-minute temporal resolution from 2009 to 2016
at the seven SURFRAD stations were used.

The second network used was NEON (Metzger et al., 2019), and 42 terrestrial

tower stations (denoted by red triangles in Fig. 1) in the network were used in this study.
Generally, measurements of the PAR vertical profile at multiple vertical levels were
conducted at each tower station and the tower-top PAR measurements were used to
validate our global gridded PAR product. Kipp & Zonen PQS 1 quantum sensors with
an uncertainty within 4% (Blonquist and Johns, 2018) were used to measure PAR across
the NEON. The sensors sampled with frequency of 1 Hz, recorded PAR values every
minute, and were calibrated every year. The starting times of PAR observations at the
42 NEON stations are different to each other, and thus here we used PAR observations
from the starting time of each site to the end of 2018.

The third network used was CERN, and 38 stations (marked with red circles in Fig.

1) across diverse terrestrial ecosystems were used in this study. These 38 CERN stations
were distributed across different climatic zones and belonged to eight different
ecosystems: agriculture, forest, desert, marine, grassland, lake, marsh wetland, and
urban. LI-190SA quantum sensors with an uncertainty of approximately 5% (Hu et al.,
2007) were used to measure PAR across CERN, and the spectrometer and standard
radiative lamp were adopted to centralized calibrate and compare among the quantum
sensors. The PAR observations were recorded hourly and thus we only validated our
daily PAR product against CERN due to the mismatch between the hourly observed
data and the satellite-based instantaneous retrievals. The daily mean PAR datasets from
the 38 CERN stations from 2005 to 2015 were publicly shared by Liu et al. (2017) and
used herein.





**4 Results and Discussion**

Based on the above inputs and the physical-based PAR algorithm, we produced a long-term (from 1984 to 2018) high resolution (10 km spatial resolution and 3 hours temporal resolution) global gridded PAR product, here referred to as the ISCCP-ITP PAR product. In-situ observations from three networks were used to evaluate the performance of our ISCCP-ITP PAR product at instantaneous and daily scales. In addition, a widely used global gridded PAR product of the CERES (SYN1deg-1hour, edition 4A), with a spatial resolution of $1^\circ \times 1^\circ$ and a temporal resolution of 1 hour, was used to provide a comparison with our ISCCP-ITP PAR product. To discuss the influence of spatial resolution on the accuracy of our global gridded PAR product, we also evaluated the estimated PAR at different spatial resolutions from 10 km to 110 km. The estimated PAR at spatial resolutions from 30 km to 110 km were calculated by averaging the corresponding original PAR at the 10 km scale. Here, the three statistical metrics of mean bias error (MBE), RMSE, and correlation coefficient ($R$), were used to evaluate the performance of our ISCCP-ITP PAR product and the CERES PAR product.

**4.1 Validation of instantaneous PAR**

In this study, the instantaneous PAR was validated against the observed hourly PAR, which was calculated by averaging the 1-minute PAR over the time period of 30 minutes before and after satellite overpass. Our estimated instantaneous PAR was firstly validated against in-situ data measured at the seven SURFRAD stations. Figure 2 presents the validation results for the instantaneous PAR at spatial resolutions of 10 km and 30 km, and the validation result for the CERES hourly PAR with a spatial resolution of approximately 100 km. It can be seen that the accuracy of the instantaneous PAR at 10 km spatial resolution (MBE = 5.6 W m$^{-2}$, RMSE = 44.3 W m$^{-2}$, $R$ = 0.94) is

comparable to that of the CERES hourly PAR at 100 km spatial resolution (MBE = 4.9
W m$^{-2}$, RMSE = 44.1 W m$^{-2}$, $R$ = 0.93). However, when the instantaneous PAR at 10
km spatial resolution was averaged to 30 km, its accuracy was markedly improved;
RMSE decreased from 44.3 to 36.3 W m$^{-2}$ and $R$ increased from 0.94 to 0.96, and thus
its accuracy at 30 km spatial resolution is clearly higher than that of the CERES product.

Table 1 shows the accuracies of our estimated instantaneous PAR at different

spatial resolutions from 10 km to 110 km. It can be seen that the accuracy at the original
10 km spatial resolution was clearly lower than at all other resolutions (30–110 km),
and the accuracy was highest at a resolution of 50–70 km. This may be due to the
following two reasons. Firstly, the representativeness of ground-based observational
stations may be greater than 10 km. Secondly, there is time mismatch between satellite-
based and surface-based observations because the last generation of geostationary
meteorological satellites (e.g., the Geostationary Operational Environmental Satellite
(GOES)) require approximately half an hour to complete a disk scan. Spatially
averaging the instantaneous PAR to a larger area could partially eliminate this time
mismatch.

The instantaneous PAR was also evaluated against the 42 NEON stations (Figure

3 and Table 2). The performance against NEON was slightly worse than that against
SURFRAD. At the 10 km scale, the former produced a 1.2 W m$^{-2}$ larger RMSE than the
latter, and both produced a positive MBE of approximately 6 W m$^{-2}$ and $R$ of 0.94.
Similar to the situation at SURFRAD, the accuracy at NEON was markedly improved
at 30 km spatial resolution, reached a peak at 50 km resolution, and then started to
decrease slightly at 70 km resolution. Compared to the performance of the CERES
hourly PAR at NEON, the accuracy of our estimated instantaneous PAR was higher at
all scales from 10 km to 110 km. More importantly, the spatial resolution of our PAR



product (10 km) is much finer than that of the CERES PAR product (100 km).
Due to the significant improvement when our estimated PAR was upscaled to 30
km spatial resolution, we used a 3 × 3 spatial window to smooth the raw PAR to derive
our final global grided PAR product. Thus, we here present the spatial distributions of
MBE and RMSE (Figure 4) for our estimated PAR with a spatial resolution of 30 km
across seven SURFRAD and 42 NEON stations in the USA. The MBE values range
from −11.2 to 19.8 W m$^{-2}$, with a negative MBE at 5 of the 49 stations. From an MBE
point of view, 42 stations fall into the range −10 to 10 W m$^{-2}$, and among these 22
stations fall within −5 to 5 W m$^{-2}$. The RMSE values range from 24.2 to 52.3 W m$^{-2}$,
with RMSE ≤ 35 W m$^{-2}$ at 18 stations, RMSE between 35 and 40 W m$^{-2}$ at 19 stations,
RMSE between 40 and 50 W m$^{-2}$ at 12 stations, and RMSE > 50 W m$^{-2}$ at only one
station. The largest MBE and RMSE both occur at the Great Smoky Mountains National
Park (GRSM) station, which is situated in the mountains of southeastern Tennessee.
Similar large errors at this station were also found for the CERES PAR product. The
relatively large errors at this station could be caused by the poor representativeness of
the mountain observational station.

**4.2 Validation of daily PAR**
Our estimated daily PAR (ISCCP-ITP) was derived by averaging the instantaneous
PAR of eight moments in the day, and validated against the three networks of
SURFRAD, NEON, and CERN. Similar to the validation results for the instantaneous
PAR, the performance of our estimated daily PAR at 10 km spatial resolution was
comparable to that of the CERES product at SURFRAD and NEON, and when upscaled
to ≥ 30 km, our daily PAR product performed slightly better than that of CERES.
Therefore, here we do not give validation results for the CERES daily PAR at
SURFRAD and NEON, but only give validation results for the CERES daily PAR at
CERN.

Validation results for our estimated daily PAR against in-situ data collected at

SURFRAD are shown in Figure 5 and Table 3. The MBE, RMSE, and $R$ values were
0.4 W m$^{-2}$, 13.2 W m$^{-2}$, and 0.96, respectively, for daily PAR at 10 km spatial resolution.
When upscaled to 30 km spatial resolution, these statistical metrics changed to 0.6 W
m$^{-2}$, 11.2 W m$^{-2}$, and 0.97, respectively. When upscaled to ≥50 km, the RMSE gradually
decreased to approximately 10 W m$^{-2}$. The MBE and $R$ changed to 0.5 W m$^{-2}$ and 0.98,
respectively.

Validation results for our estimated daily PAR against NEON are shown in Figure

6 and Table 4. The RMSE for daily PAR at 10 km spatial resolution was 13.1 W m$^{-2}$,
and this value decreased to 11.6 W m$^{-2}$ for 30 km spatial resolution. The $R$ for daily
PAR was 0.96 and 0.97 for 10 km and 30 km spatial resolution, respectively. When
upscaled to ≥ 50 km, these statistical metrics remained almost unchanged. The
performance against NEON is comparable to that against SURFRAD for our daily PAR
product.

Figure 7 shows the spatial distributions of MBE and RMSE for our estimated daily

PAR with a spatial resolution of 30 km against seven SURFRAD and 42 NEON stations
in the USA. The largest negative and positive MBE values were −5.3 W m$^{-2}$ and 9.3 W
m$^{-2}$, respectively. There were seven stations with MBE < 0 W m$^{-2}$, 41 stations with
MBE values between −5 W m$^{-2}$ and 5 W m$^{-2}$, 31 stations with MBE values between −3
W m$^{-2}$ and 3 W m$^{-2}$, and only eight stations with absolute MBE > 5 W m$^{-2}$. The largest
and smallest RMSE values were 17.6 W m$^{-2}$, and 6.9 W m$^{-2}$, respectively. There were
12 stations with RMSE < 10 W m$^{-2}$, 19 stations with RMSE between 10 W m$^{-2}$ and 12
W m$^{-2}$, 12 stations with RMSE between 12 W m$^{-2}$ and 13 W m$^{-2}$, and only six stations



with RMSE > 13 W m⁻². Likewise, the largest MBE and RMSE values were found at
the GRSM station with the main reason again likely being due to the poor
representativeness of this station.

Finally, we validated our daily PAR and the CERES daily PAR products against

in-situ data collected across CERN (Figure 8). The performance of our daily PAR
product at the 10 km scale (MBE = 1.4 W m⁻², RMSE = 19.6 W m⁻², $R$ = 0.89) was
slightly worse than that of the CERES daily PAR product (MBE = −1.3 W m⁻², RMSE
= 18.7 W m⁻², $R$ = 0.90). However, when upscaled to ≥ 30 km, the accuracies of our
estimated daily PAR were comparable to, or slightly better than, those of the CERES
daily PAR. Another phenomenon we noticed was that the RMSEs against CERN data
were approximately 7−8 W m⁻² greater than those against SURFRAD and NEON data
for both our daily PAR and the CERES PAR products. This could be attributed to the
fact that the quality of PAR observations at CERN is slightly worse than that at
SURFRAD and NEON, but further evidence is required to support this speculation.
Another possible reason could be the effect of aerosols because aerosols are a major
attenuation factor affecting the clear-sky PAR (Qin et al., 2012; Tang et al. 2013).
Because the aerosol optical depth (AOD) over China is much greater than that over the
USA (Li et al., 2011), greater uncertainty in the aerosol data over China would lead to
larger errors in PAR estimation over China.

Figure 9 presents the spatial distributions of MBE and RMSE for our estimated

daily PAR with a spatial resolution of 30 km against the 38 CERN stations. The MBE
values at most of the stations were between −10 W m⁻² and 10 W m⁻². The stations with
negative MBE were mainly located in northwestern China, and the stations with
positive MBE were mainly located in southeastern China. The RMSE values at most of
the stations were < 23 W m⁻², and there were only five stations where the RMSE was >



25 W m$^{-2}$. Stations with an absolute MBE > 10 W m$^{-2}$ were mainly located in four
forested areas (Beijing, Xishuangbanna, Heshan, and Ailao Mountain), one agricultural
area (Huanjiang), one lake area (Taihu), and one Desert area (Fukang). Likewise, the
RMSE values at these seven stations were relatively large. Similar large errors at these
stations were also found for the CERES PAR product. The large errors at these stations
could be caused by the poor representativeness at some mountain stations, large
uncertainty in the inputs at some stations, or uncertainty in observational data.

### 4.3 Spatial distribution of multi-year average PAR

Figure 10 shows the global spatial distribution of multi-year annual average PAR
(ISCCP-ITP) during the period 2001–2018, and comparison with that of the CERES
PAR is also shown. The spatial pattern of our ISCCP-ITP PAR product is quite
consistent with that of the CERES PAR product, whose spatial resolution was far
coarser than that of our PAR product. There were some finer patterns that the CERES
PAR product could not distinguish, but our PAR product could clearly capture. This
defect in the CERES PAR product was especially evident in mountainous areas, such
as the Tibetan Plateau. The annual average PAR was generally high in latitudinal zones
lying between 30º N and 30º S, and low in other regions. In addition, there were some
high-altitude regions with high PAR values, such as the Tibetan Plateau and Bolivian
Plateau.
Figure 11 displays the global spatial distributions of multi-year seasonal average
PAR (ISCCP-ITP) during the period 2001–2018. The four panels in the figure reflect
the process of seasonal change and exhibit different spatial distribution characteristics.
Compared to mid- and high-latitude areas, more PAR was received around the equator
and low latitudes (30º N-30º S) in all four seasons. Over the latitudinal zone between



30º S and 90 º S in southern hemisphere, PAR received by the surface gradually
increased from spring to winter, with the lowest values in spring and summer, a
relatively larger value in autumn, and the largest value in winter. Over the latitudinal
zone between 30º N and 90 º N in northern hemisphere, the situation was very different.
PAR received by the surface was largest in summer, lowest in autumn and winter, and
intermediate in spring.

**5 Data availability**
Our long-term global gridded PAR product is available at the National Tibetan
Plateau Data Center (https://doi.org/10.11888/RemoteSen.tpdc.271909, Tang, 2021),
Institute of Tibetan Plateau Research, Chinese Academy of Sciences.

**6 Summary and Conclusions**
A long-term (1984–2018) global high-resolution (10 km spatial resolution, 3 h
temporal resolution) gridded PAR product was produced using our previously published
physical-based PAR parametrization scheme. The main inputs for this PAR model were
the latest ISCCP H-series cloud product, ERA5 routine meteorological data (water
vapor, surface pressure, and ozone), MERRA-2 aerosol product, and albedo products
from MODIS (after 2000) and CLARRA-2 (before 2000). The generated PAR product
was validated globally against in-situ data measured across three observational
networks in the USA and China. For the instantaneous PAR at original the scale (10
km), the overall MBE, RMSE, and R were 5.8 W m$^{-2}$, 44.9 W m$^{-2}$ and 0.94, respectively.
When smoothed to $\geq$ 30 km, the accuracy was markedly improved, with RMSE
decreasing to 37.1 W m$^{-2}$ and R increasing to 0.96. For the daily PAR at spatial
resolutions of 10 km and 30 km, the RMSE values were approximately 13.1 W m$^{-2}$ and



11.4 W m$^{-2}$, respectively, in the USA. Validation results in China showed a greater
RMSE than in the USA. Due to the marked improvement when our PAR products were
upscaled to ≥ 30 km, we applied a 3×3 spatial smoothing window to the original PAR
data to produce the final PAR product.

Our estimated PAR product was also compared with the CERES PAR product; we

found that the accuracy of our estimated PAR product at the original scale (10 km) was
generally comparable to, or higher than, that of the CERES PAR product. When it was
upscaled to ≥ 30 km, the accuracy advantage of our product over the CERES PAR
product became more evident. Another clear advantage of our PAR product was the
increased spatial resolution it offered compared to the CERES PAR product. We expect
that our PAR product will contribute to the future understanding and modeling of the
global carbon cycle and ecological processes. In future work, we will attempt to
separate the components of direct and diffuse PAR from the total PAR because light use
efficiency is mainly controlled by diffuse PAR.

**Author contributions.** All authors discussed the results and contributed to the
manuscript. WT calculated the dataset, analyzed the results, and drafted the manuscript.

**Competing interests.** The authors declare that they have no conflicts of interest.

**Acknowledgments.** The in-situ observations of PAR at CERN were shared by Liu et
al. (2017) and are available online via http://www.sciencedb.cn/dataSet/handle/326.
The observed PAR data at SURFRAD and NEON are available online from their
official        websites        (https://www.esrl.noaa.gov/gmd/grad/surfrad/       and
http://data.neonscience.org). The ISCCP H-series cloud products were provided by the



NOAA's National Centers for Environmental Information (NCEI). The ERA5 routine
weather data, MODIS albedo data, and MERRA-2 aerosol data are available from their
official websites (https://www.ecmwf.int, https://ladsweb.modaps.eosdis.nasa.gov, and
https://gmao.gsfc.nasa.gov/reanalysis/MERRA-2/). The authors would like to thank the
staff members at these observational networks and data production centers for their
valuable work.

**Financial support.** This work was supported by the National Key Research and
Development Program of China (Grant No. 2017YFA0603604), and the National
Natural Science Foundation of China (Grant No. 42171360).

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



**Figure captions**

**Figure 1** Distribution of observation stations within the three observation networks, where measurements of PAR were carried out. The red circles denote the locations of the 38 CERN stations, the red triangles denote the 42 NEON stations, and the red pentagrams denote the seven SURFRAD stations.

**Figure 2** Comparisons of our estimated instantaneous PAR product (ISCCP-ITP) at spatial resolutions of (a) 10 km, (b) 30 km, and (c) hourly PAR of the CERES SYN1deg (edition 4.1) with observed PAR collected at seven SURFRAD stations.

**Figure 3** Comparisons of our estimated instantaneous PAR product (ISCCP-ITP) at spatial resolutions of (a) 10 km, (b) 30 km, and (c) hourly PAR of the CERES SYN1deg (edition 4.1) with observed PAR collected at 42 NEON stations.

**Figure 4** Spatial distribution of (a) MBE (W m$^{-2}$) and (b) RMSE (W m$^{-2}$) for our estimated instantaneous PAR product (ISCCP-ITP, 30 km) at seven SURFRAD stations and 42 NEON stations.

**Figure 5** Comparisons of our estimated daily PAR product (ISCCP-ITP) at spatial resolutions of (a) 10 km and (b) 30 km with observed PAR collected at seven SURFRAD stations.

**Figure 6** Comparisons of our estimated daily PAR product (ISCCP-ITP) at spatial resolutions of (a) 10 km and (b) 30 km with observed PAR collected at 42 NEON stations.

**Figure 7** Same as **Figure 4, but for** our estimated daily PAR product (ISCCP-ITP, 30 km).

**Figure 8** Comparisons of our estimated daily PAR product (ISCCP-ITP) at spatial resolutions of (a) 10 km, (b) 30 km, and (c) daily PAR of the CERES

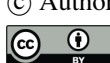



SYN1deg (edition 4.1) with observed PAR collected at 38 CERN stations.

**Figure 9** Spatial distribution of (a) MBE (W m$^{-2}$) and (b) RMSE (W m$^{-2}$) for our

estimated daily PAR product (ISCCP-ITP, 30 km) at 38 CERN stations.

**Figure 10** Spatial distribution of annual mean PAR between 2001 and 2018, derived

from (a) our estimated PAR product (ISCCP-ITP), and (b) the CERES PAR

product. The unit of PAR is W m$^{-2}$.

**Figure 11** Spatial distribution of seasonal mean PAR between 2001 and 2018 derived

from our estimated PAR product (ISCCP-ITP). The unit of PAR is W m$^{-2}$.

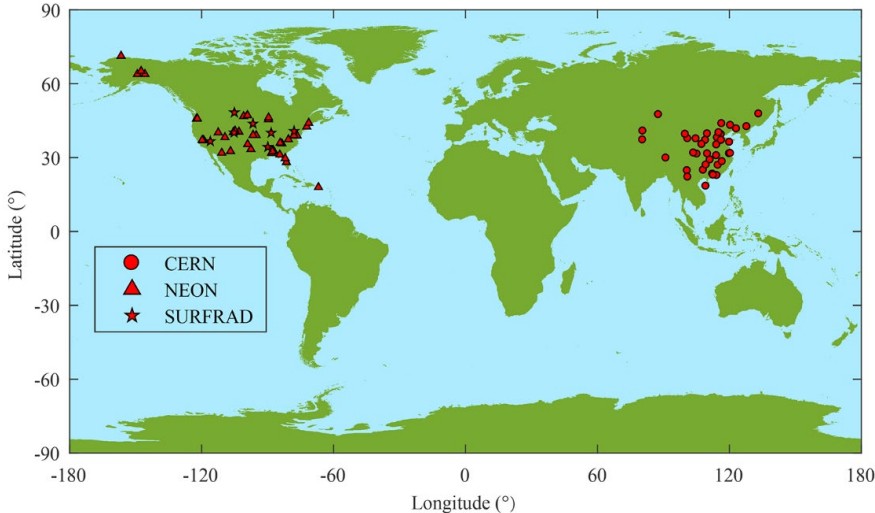


**Figure 1** Distribution of observation stations within the three observation networks,

where measurements of PAR were carried out. The red circles denote the

locations of the 38 CERN stations, the red triangles denote the 42 NEON

stations, and the red pentagrams denote the seven SURFRAD stations.

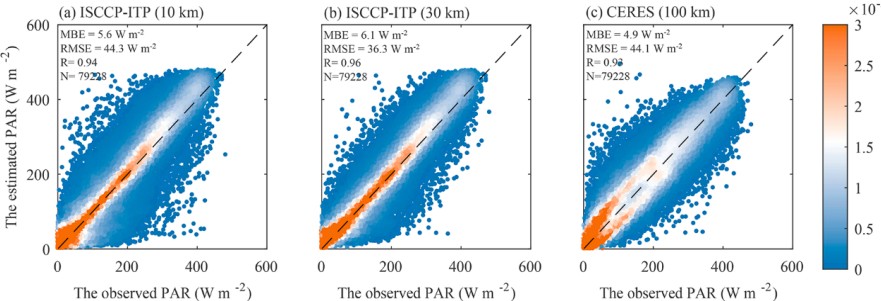

**Figure 2** Comparisons of our estimated instantaneous PAR product (ISCCP-ITP) at spatial resolutions of (a) 10 km, (b) 30 km, and (c) hourly PAR of the CERES SYN1deg (edition 4.1) with observed PAR collected at seven SURFRAD stations.

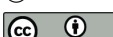

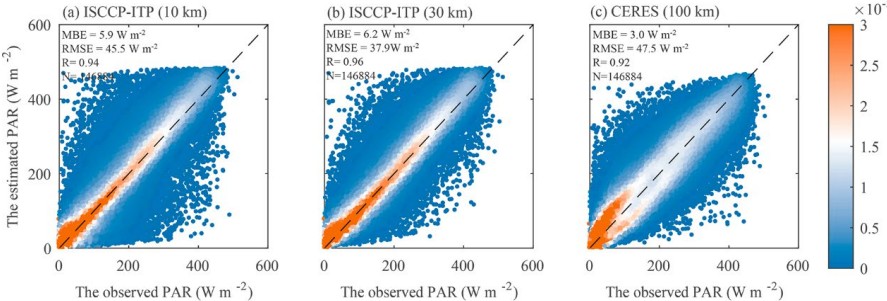


**Figure 3** Comparisons of our estimated instantaneous PAR product (ISCCP-ITP) at

spatial resolutions of (a) 10 km, (b) 30 km, and (c) hourly PAR of the CERES

SYN1deg (edition 4.1) with observed PAR collected at 42 NEON stations.



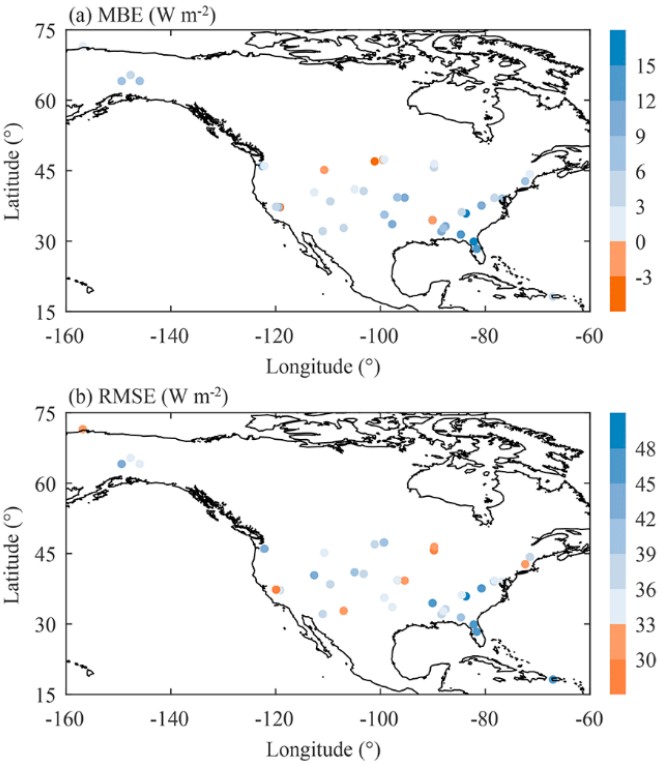


**Figure 4** Spatial distribution of (a) MBE (W m$^{-2}$) and (b) RMSE (W m$^{-2}$) for our estimated instantaneous PAR product (ISCCP-ITP, 30 km) at seven SURFRAD stations and 42 NEON stations.


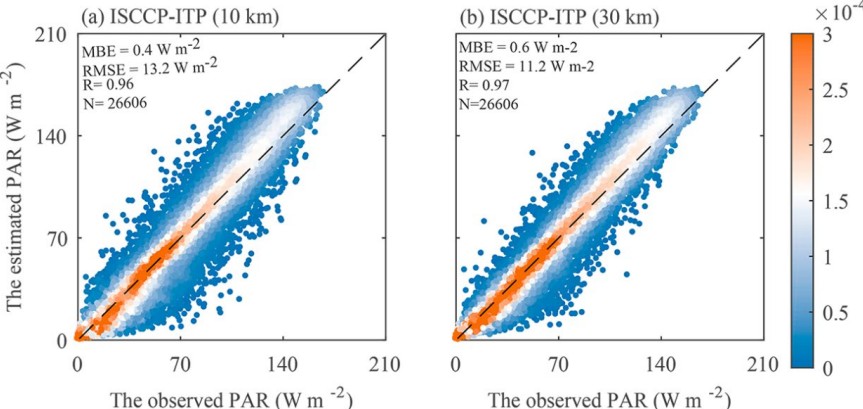

**Figure 5** Comparisons of our estimated daily PAR product (ISCCP-ITP) at spatial resolutions of (a) 10 km and (b) 30 km with observed PAR collected at seven SURFRAD stations.

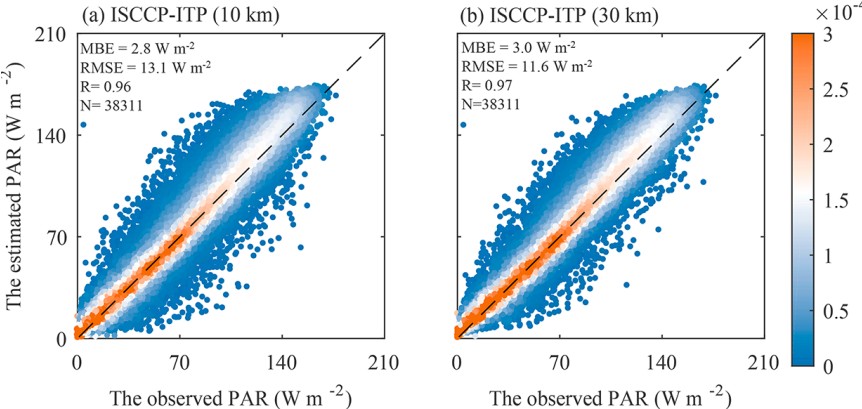

**Figure 6** Comparisons of our estimated daily PAR product (ISCCP-ITP) at spatial resolutions of (a) 10 km and (b) 30 km with observed PAR collected at 42 NEON stations.

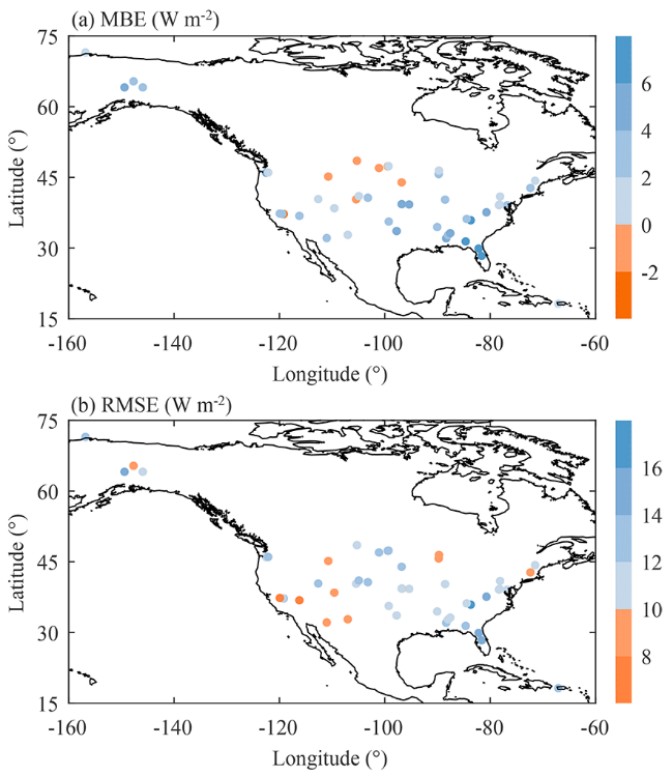

**Figure 7** Same as **Figure 4**, but for our estimated daily PAR product (ISCCP-ITP, 30 km).

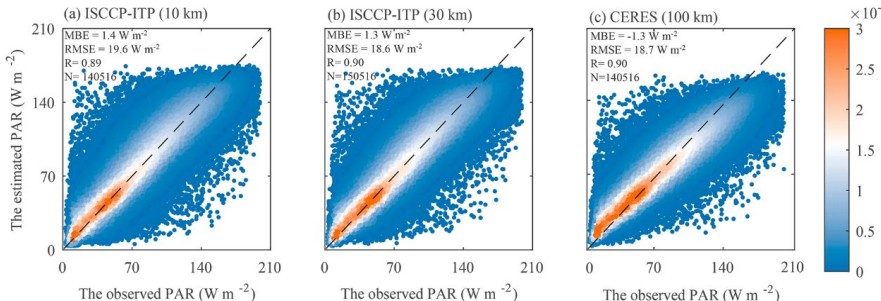

**Figure 8** Comparisons of our estimated daily PAR product (ISCCP-ITP) at spatial resolutions of (a) 10 km, (b) 30 km, and (c) daily PAR of the CERES SYN1deg (edition 4.1) with observed PAR collected at 38 CERN stations.
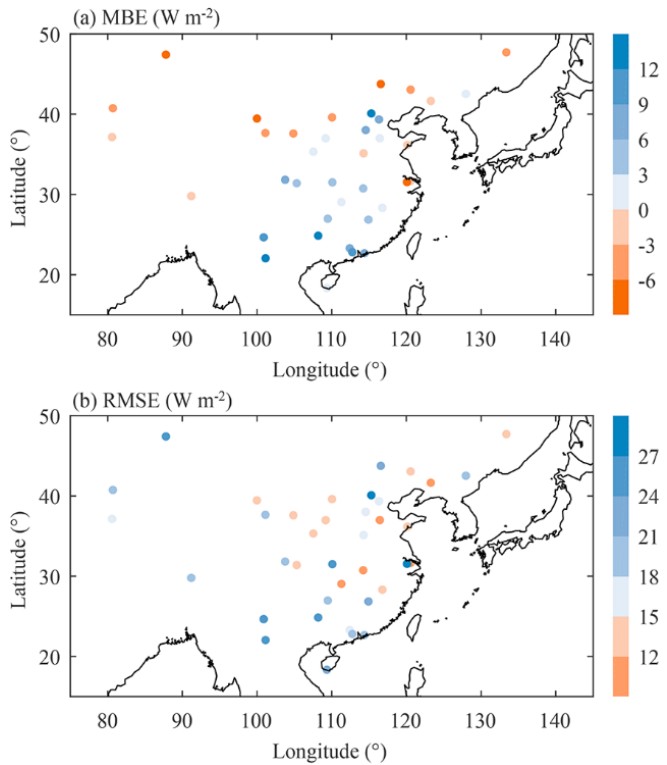

**Figure 9** Spatial distribution of (a) MBE (W m$^{-2}$) and (b) RMSE (W m$^{-2}$) for our

estimated daily PAR product (ISCCP-ITP, 30 km) at 38 CERN stations.

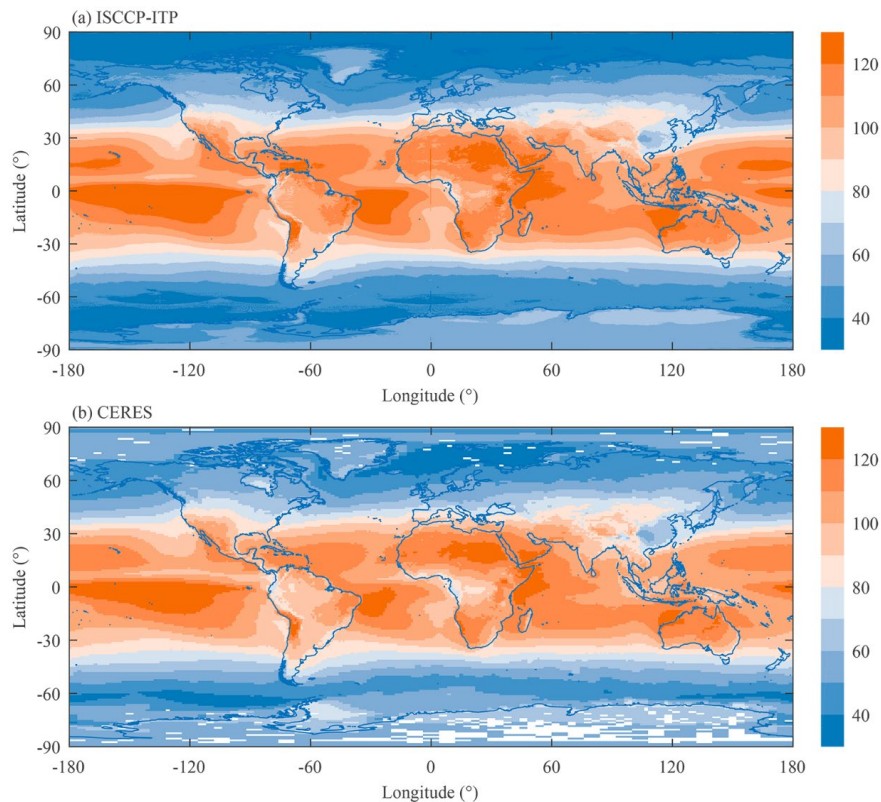

**Figure 10** Spatial distribution of annual mean PAR between 2001 and 2018, derived

from (a) our estimated PAR product (ISCCP-ITP), and (b) the CERES PAR

product. The unit of PAR is W m$^{-2}$.





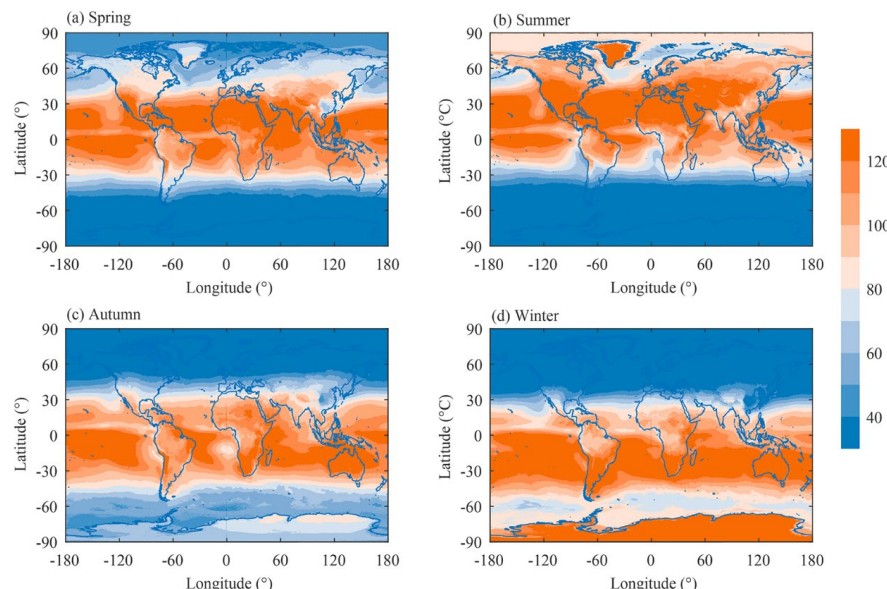

**Figure 11** Spatial distribution of seasonal mean PAR between 2001 and 2018 derived

from our estimated PAR product (ISCCP-ITP). The unit of PAR is W m⁻².



**Table captions**

**Table 1**. Effect of spatial resolution (from 10 km to 110 km) on accuracy of our estimated instantaneous PAR product (ISCCP-ITP) compared to observations at the seven SURFRAD stations.

**Table 2**. Effect of spatial resolution (from 10 km to 110 km) on accuracy of our estimated instantaneous PAR product (ISCCP-ITP) compared to observations at the 42 NEON stations.

**Table 3**. Effect of spatial resolution (from 10 km to 110 km) on accuracy of our estimated daily PAR product (ISCCP-ITP) compared to observations at the seven SURFRAD stations.

**Table 4**. Effect of spatial resolution (from 10 km to 110 km) on accuracy of our estimated daily PAR product (ISCCP-ITP) compared to observations at the 42 NEON stations.

**Table 5**. Effect of spatial resolution (from 10 km to 110 km) on accuracy of our estimated daily PAR product (ISCCP-ITP) compared to observations at the 38 CERN stations.



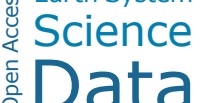

**Table 1**. Effect of spatial resolution (from 10 km to 110 km) on accuracy of our

estimated instantaneous PAR product (ISCCP-ITP) compared to observations

at the seven SURFRAD stations.

|  | Spatial resolution | MBE (W m$^{-2}$) | RMSE (W m$^{-2}$) | $R$ |
|---|---|---|---|---|
| ISCCP-ITP | 10 km | 5.6 | 44.3 | 0.94 |
| ISCCP-ITP | 30 km | 6.1 | 36.3 | 0.96 |
| ISCCP-ITP | 50 km | 6.0 | 35.0 | 0.96 |
| ISCCP-ITP | 70 km | 5.9 | 35.1 | 0.96 |
| ISCCP-ITP | 90 km | 6.0 | 35.5 | 0.96 |
| ISCCP-ITP | 110 km | 5.9 | 36.0 | 0.96 |






**Table 2**. Effect of spatial resolution (from 10 km to 110 km) on accuracy of our
estimated instantaneous PAR product (ISCCP-ITP) compared to observations
at the 42 NEON stations.

| | Spatial resolution | MBE (W m$^{-2}$) | RMSE (W m$^{-2}$) | $R$ |
|---|---|---|---|---|
| ISCCP-ITP | 10 km | 5.9 | 45.5 | 0.94 |
| ISCCP-ITP | 30 km | 6.2 | 37.9 | 0.96 |
| ISCCP-ITP | 50 km | 6.3 | 37.0 | 0.96 |
| ISCCP-ITP | 70 km | 6.2 | 37.4 | 0.96 |
| ISCCP-ITP | 90 km | 6.2 | 38.0 | 0.96 |
| ISCCP-ITP | 110 km | 6.1 | 38.6 | 0.95 |



**Table 3**. Effect of spatial resolution (from 10 km to 110 km) on accuracy of our

estimated daily PAR product (ISCCP-ITP) compared to observations at the

seven SURFRAD stations.

|          | Spatial resolution | MBE (W m$^{-2}$) | RMSE (W m$^{-2}$) | $R$ |
|----------|--------------------|------------------|-------------------|------|
| ISCCP-ITP | 10 km   | 0.4 | 13.2 | 0.96 |
| ISCCP-ITP | 30 km   | 0.6 | 11.2 | 0.97 |
| ISCCP-ITP | 50 km   | 0.5 | 10.5 | 0.98 |
| ISCCP-ITP | 70 km   | 0.5 | 10.1 | 0.98 |
| ISCCP-ITP | 90 km   | 0.5 | 9.9  | 0.98 |
| ISCCP-ITP | 110 km  | 0.5 | 9.8  | 0.98 |







**Table 4**. Effect of spatial resolution (from 10 km to 110 km) on accuracy of our

estimated daily PAR product (ISCCP-ITP) compared to observations at the

42 NEON stations.

|  | Spatial resolution | MBE (W m$^{-2}$) | RMSE (W m$^{-2}$) | $R$ |
|---|---|---|---|---|
| ISCCP-ITP | 10 km | 2.8 | 13.1 | 0.96 |
| ISCCP-ITP | 30 km | 3.0 | 11.6 | 0.97 |
| ISCCP-ITP | 50 km | 3.0 | 11.4 | 0.97 |
| ISCCP-ITP | 70 km | 3.0 | 11.5 | 0.97 |
| ISCCP-ITP | 90 km | 3.0 | 11.7 | 0.97 |
| ISCCP-ITP | 110 km | 2.9 | 11.8 | 0.97 |






**Table 5**. Effect of spatial resolution (from 10 km to 110 km) on accuracy of our
estimated daily PAR product (ISCCP-ITP) compared to observations at the
38 CERN stations.

| | Spatial resolution | MBE (W m$^{-2}$) | RMSE (W m$^{-2}$) | $R$ |
|---|---|---|---|---|
| ISCCP-ITP | 10 km | 1.4 | 19.6 | 0.89 |
| ISCCP-ITP | 30 km | 1.3 | 18.6 | 0.90 |
| ISCCP-ITP | 50 km | 1.2 | 18.3 | 0.90 |
| ISCCP-ITP | 70 km | 1.2 | 18.3 | 0.90 |
| ISCCP-ITP | 90 km | 1.1 | 18.2 | 0.90 |
| ISCCP-ITP | 110 km | 1.1 | 18.3 | 0.90 |
