# Peer review of "Mapping long-term and high-resolution global gridded photosynthetically active"

_Earth System Science Data, 2022_

## Author Comment (AC1)

**Response to Referee #1**

We would like to thank the reviewer for the comments and suggestions, which help to improve the quality of our work. We have made revisions and have replied to all comments and suggestions. Please find a detailed point-by-point response to each comment.

**Comment:**

This manuscript produced a long-term and high-resolution global gridded PAR product based on the latest ISCCP-HXG cloud product and reanalysis data. PAR data is required for researches in the ecological, agricultural, and global change fields. The algorithm used to estimate PAR in this study is the physical scheme that used in the previous study of Tang et al. (2017), and was proven to be a more accurate algorithm than previous ones. The produced gridded PAR product was evaluated against surface observations collected at more than 80 experimental stations worldwide. Compared with the well-known CERES PAR product, the PAR product produced in this study was found to be a more accurate dataset with higher resolution. The topic is highly interesting and appropriated for ESSD. The paper is clear and well written. Therefore, I recommend its publishing on the ESSD after answering the following several minor issues.

**Response:**

We thank Referee #1 for the encouraging comments. All comments and suggestions have been considered carefully and well addressed.

**Comment:**

1. In this study, aerosol data from the reanalysis data MERRA-2 was used? Why not use the satellite-based aerosol products with higher accuracy?

**Response:**

Although satellite retrieved aerosol products may be more accurate than other aerosol products, there are some defects in satellite retrieved aerosol products, such as many missing values when clouds exist, no observations before 2000.

Alternatively, the MERRA-2 can provide long-term gridded aerosol product. In addition, the accuracy of MERRA-2 aerosol product is comparable to those of satellite-retrieved aerosol products because it assimilates ground-observed and satellite-retrieved aerosol data. Therefore, the MERRA-2 aerosol product is more suitable than satellite-based aerosol product for this study.

**Comment:**

2. Cloud top temperature was used to discriminate the water and ice cloud in this study. Since MODIS has a cloud top temperature product, and why not use this product?

**Response:**

Yes, MODIS has a cloud top temperature product. It would be valuable to use the MODIS cloud top temperature product since it was considered to be the benchmark for cloud product, but its temporal resolution is too low and there was no product before 2000. In addition, the big mismatches between the times of MODIS and ISCCP H-series cloud product will introduce great uncertainty.

**Comment:**

3. In section of in-situ measurements, seven experimental stations from SURFRAD, 42 experimental stations from NEON, and 38 experimental stations from CERN were used to evaluate the performance of the estimated PAR. Did you do quality control on these observations and what are the criteria for control?

**Response:**

The PAR observations collected at the SURFRAD and NEON networks used in this study were quality controlled by station scientists before release, and are regarded as the most reliable radiation data due to the instruments of highest available accuracy and careful maintenance.

The PAR observations collected at the CERN network used in this study were quality controlled by the data sharers, more details about the quality control procedure can be found in the article of Liu et al. (2017). This information will be added in the revised manuscript as "The PAR observations collected at the CERN network were quality controlled by the data sharers, more details about the quality control procedure can be found in the article of Liu et al. (2017)".

---

## Author Comment (AC2)

**Response to Referee #2**

We would like to thank the reviewer for the comments and suggestions, which help to improve the quality of our work. We have made revisions and have replied to all comments and suggestions. Please find a detailed point-by-point response to each comment.

**Comment:**

The paper by Tang et al. produced a 35-year (1984-2018) high-resolution (3 h, 10 km) global gridded PAR dataset using ISCCP, MERRA-2, ERA5, MODIS and CLARRA-2 products as inputs in a physical-based model. In the paper, authors compared their instantaneous and daily PAR products against surface experimental stations, including SURFRAD, NEON, CERN networks, and CERES products. The results prove that the PAR product was found to be a more accurate dataset with higher resolution. This study could be a good supplement to meet refined analysis and understand climate variations. I believe this study can be valuable to the relevant community.

Overall, the study describes the background and introduction, and methods in a comprehensive way. However, the method in the paper contains inadequate innovations. Therefore, I would encourage the authors to submit a revised manuscript by addressing my specific comments below:

**Response:**

We thank Referee #2 for the encouraging comments. All comments and suggestions have been considered carefully and well addressed.

**Comment:**

1. First of all, to ensure the accuracy of surface network data, Does the surface network data used in the paper undergo the data preprocessing and the removal process of invalid values.

**Response:**

We did not perform any quality control on these observations because they were done before release. The PAR observations collected at the SURFRAD and NEON networks used in this study were quality controlled by station scientists before release, and are regarded as the most reliable radiation data due to the instruments of highest available accuracy and careful maintenance.

The PAR observations collected at the CERN network used in this study were quality controlled by the data sharers, more details about the quality control procedure can be found in the article of Liu et al. (2017). This information will be added in the revised

manuscript as "The PAR observations collected at the CERN network were quality controlled by the data sharers, more details about the quality control procedure can be found in the article of Liu et al. (2017)".

Liu, H., et al., (2017), CERN photosynthetically active radiation dataset from 2005 to 2015, China Scientific Data, 2017, 2(1), 1–10. DOI: 10.11922/csdata.170.2016.0100.

**Comment:**

2. How to compare ground-based stations with satellite pixels, the comparison process could introduce errors in the results.

**Response:**

Here, we directly compared the ground-based observations with the estimated PAR values of the corresponding satellite pixel. Yes, the comparison process would introduce some uncertainty in the results. This is also an issue of site representativeness. If a site is representative of the corresponding satellite pixel, then the uncertainty in the validation result is negligible, otherwise the uncertainty is non-negligible. Generally, the representativeness of a site over flat area can greater than 25 km for downward shortwave radiation according to Schwarz et al. (2017) and Huang et al. (2019). In this study, most of the experimental stations are over flat areas, and thus the uncertainty in the validation result of this study is negligible. In addition, to further discuss the issue of site representativeness on the accuracy of our global gridded PAR product, we also evaluated the estimated PAR at different spatial resolutions from 10 km to 110 km.

In response to your issue, part of the above information will be added into the revised manuscript as "Here, we directly compared the ground-based observations with the estimated PAR values of the corresponding satellite pixel".

Huang, G., Li, Z., Li, X., Liang, S., Yang, K., Wang, D., and Zhang, Y., 2019: Estimating surface solar irradiance from satellites: Past, present, and future perspectives. Remote Sensing of Environment, 233, 111371.

Schwarz, M., Folini, D., Hakuba, M. Z., & Wild, M. (2017). Spatial representativeness of surface-measured variations of downward solar radiation. Journal of Geophysical Research: Atmospheres, 122, 13,319–13,337. https://doi.org/10.1002/2017JD027261.

**Comment:**

3. To some extent, the accuracy of the parameterized method used in the paper depends on the accuracy of the input data. The descriptions of the accuracy of the input data were insufficient.

**Response:**

Yes, I agree with you! Based on the finding of the sensitivity analysis (Tang et al., 2017), cloud and aerosol are two important variables that affect PAR estimates, and thus the accuracies of cloud and aerosol data will be added into in the revised manuscript as "The uncertainties in cloud detection and cloud property can be found in the official Climate Algorithm Theoretical Basis Document (C-ATBD, https://www.ncei.noaa.gov/pub/data/sds/cdr/CDRs/Cloud_Properties-ISCCP/AlgorithmDescription_01B-29.pdf). The accuracies of these cloud parameters in the latest ISCCP-H series are considered to be more reliable than those of cloud parameters in the previous ISCCP-D series." and "Gueymard and Yang (2020) have validated the MERRA-2 AOD product against 793 AERONET stations worldwide, and also compared with other aerosol products. It was found that the averaged RMSE for the MERRA-2 AOD at 550 nm was about 0.126, which was generally lower than those of other aerosol products.".

Gueymard, C. A., and Yang, D. (2020), Worldwide validation of CAMS and MERRA-2 reanalysis aerosol optical depth products using 15 years of AERONET observations. Atmospheric Environment, 225, 117216.

Tang, W. J., Qin, J., Yang, K., Niu, X. L., Min, M., and Liang, S. L.: An efficient algorithm for calculating photosynthetically active radiation with MODIS products, Remote Sensing of Environment, vol. 194, pp. 146-154, 2017.

**Comment:**

4. The algorithm used to map global gridded PAR in this study was the parameterization method developed by Tang et al. (2017), so the method used in the paper lack some innovations, more descriptions of Highlights are need in the paper.

**Response:**

The purpose of this paper is to provide a more accurate global grided PAR dataset with higher resolution, compared to other global gridded PAR products. The focus of this study is on the final PAR product, not the innovative approach for estimating PAR. In fact, the method used in this study is an efficient and purely physical parameterization method.

Actually, Wang et al. (2021) have compared five representative methods for estimating downward shortwave radiation, and found that the parameterization method performed best among them. This increases our confidence in estimating PAR with physical parameterization method. Therefore, we believe our PAR algorithm can work well and produce a high-quality global gridded PAR product.

In response to your issue, part of the above information will be added into the revised manuscript as "Wang et al. (2021) have compared five representative methods for estimating downward shortwave radiation, and found that the parameterization method performed best among them. This increases our confidence in estimating PAR with physical parameterization method. Therefore, we believe our PAR algorithm can work well and produce a high-quality global gridded PAR product.".

Dongdong Wang, D., Liang, S., Li, R., and Jia, A., (2021), A synergic study on estimating surface downward shortwave radiation from satellite data. Remote Sensing of Environment, 264, 112639.

**Comment:**

5. Is the band range of the PAR observations derived from the ground station consistent with the estimated PAR?

**Response:**

Yes, they are consistent. The band ranges for the observed PAR and the estimated PAR are both between 0.4 and 0.7 μm.